# What the Neuroscience and Psychology of Magic Reveal about Misinformation

Robert G. Alexander *, Stephen L. Macknik and Susana Martinez-Conde

Departments of Ophthalmology, Neurology, and Physiology & Pharmacology, State University of New York Downstate Health Sciences University, Brooklyn, NY 11203, USA
* Correspondence: Robert.Alexander@downstate.edu

**Abstract:** When we believe misinformation, we have succumbed to an illusion: our perception or interpretation of the world does not match reality. We often trust misinformation for reasons that are unrelated to an objective, critical interpretation of the available data: Key facts go unnoticed or unreported. Overwhelming information prevents the formulation of alternative explanations. Statements become more believable every time they are repeated. Events are reframed or given "spin" to mislead audiences. In magic shows, illusionists apply similar techniques to convince spectators that false and even seemingly impossible events have happened. Yet, many magicians are "honest liars", asking audiences to suspend their disbelief only during the performance, for the sole purpose of entertainment. Magic misdirection has been studied in the lab for over a century. Psychological research has sought to understand magic from a scientific perspective and to apply the tools of magic to the understanding of cognitive and perceptual processes. More recently, neuroscientific investigations have also explored the relationship between magic illusions and their underlying brain mechanisms. We propose that the insights gained from such studies can be applied to understanding the prevalence and success of misinformation. Here, we review some of the common factors in how people experience magic during a performance and are subject to misinformation in their daily lives. Considering these factors will be important in reducing misinformation and encouraging critical thinking in society.

**Keywords:** misinformation; magic; disinformation; science communication; fake news; science reporting; attentional misdirection; deception; inattentional blindness; critical thinking

## 1. Introduction

Discrediting false reports is crucial for reinforcing and maintaining society's trust in science. Unfortunately, members of the public are often barraged with incomplete, out-of-date, and biased information. Perhaps worse still is the spread of "disinformation", or misinformation that is actively disseminated with the intent to deceive others. False news, whether intentionally malicious or not, can distort people's views about important issues [1] and encourage them to engage in risky, unproductive, or harmful behaviors—such as taking unproven medications, and overreacting or underreacting to situations (e.g., ignoring safety recommendations [2]). Beyond their impact on health and safety, false news can facilitate terrorist propaganda [3], stock market manipulations [4], and a host of other malicious actions. Why do people believe false information, and—in many cases—maintain a belief in misinformation even after a story is retracted, disputed, or debunked?

Belief in misinformation might best be thought of as an *illusion*: a mismatch between factual reality and what is perceived and understood. Biases in information processing—what we see as appealing, intuitive, or believable—can alter our understanding of news headlines and social media posts. For example, cognitive biases can lead to implausible claims being interpreted as plausible or even proven (e.g., the Pizzagate conspiracy theory), for reasons unrelated to a story's content or its objective accuracy [5,6].

Here, we explore what another type of life experience, in which illusions and inaccurate beliefs are prevalent, can teach us. In a magic show, one of the performer's main objectives is to make spectators experience seemingly impossible things. To accomplish this, magicians mislead their audiences, preventing them from accurately determining or understanding the reality of the methods behind the magical effects. Psychology researchers—and more recently, neuroscientists—have studied magic for over a century, aiming to provide a cognitive and perceptual understanding of how magic works in the minds of spectators [7–17]. As a result, a great deal is known about the psychology of magic, including how observers come to question whether impossible events have occurred. This review relates those fundamental findings to how and why people mistakenly believe that misinformation is accurate.

## 2. Attention Illusions

We cannot possibly process all the information that the world presents to us. Beyond our sensory (visual, auditory, etc.) constraints, we are largely limited to noticing only those things that we attend to. Objects or events that are new, unusual, moving, or otherwise salient strongly draw our attention in a bottom-up manner (a process called "attentional capture" in psychology). When more than one event is visible, our attention tends to follow the more salient event or the one that begins first [16,18–21]. We can also direct our attention to specific objects and events in a top-down manner, as guided by our goals and intentions. Magicians manipulate the audience's attention using both bottom-up and top-down processes to control what is noticed [16,19,22].

Psychologist and magicians often think of attention as a "spotlight" that enhances the items or parts of a scene we are focusing on—see Figure 1. The attentional spotlight is not an accurate metaphor from a neuroscientific perspective, however: rather than making our object of interest more salient, our attentional system *actively suppresses* (via inhibitory neurons) how salient everything else is. That is, rather that illuminating whatever draws our interest, our attention gives us tunnel vision: it obscures all the rest [23–33]. Suppressed crucial information passes us by without ever entering our awareness.

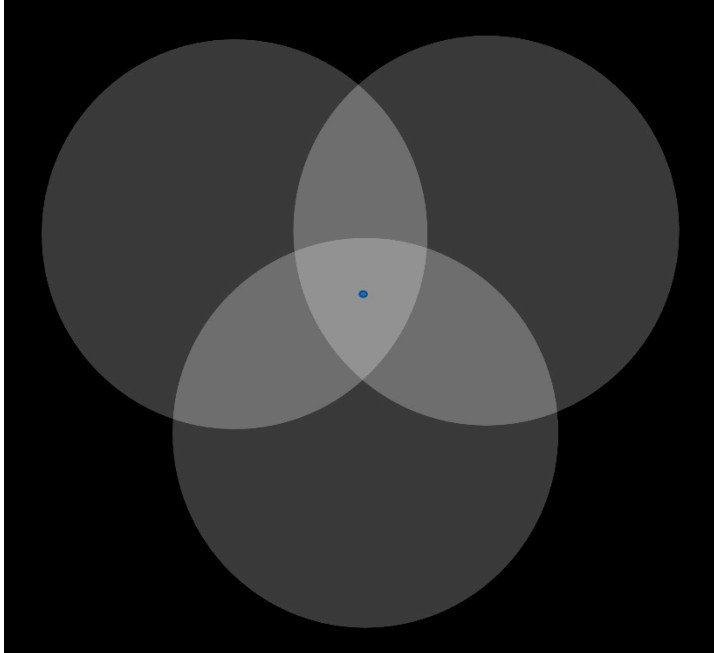

**Figure 1.** Attentional allocation modulates the perceived saliency and brightness of visual objects. Fixate your gaze on the central blue dot while directing your attention to one of the three overlapping circles. Whichever circle you focus your attention on will appear brighter than the other two (unattended) circles, and it will also seem to be positioned in front of them. Modified from [34].

### 2.1. Information Overwhelm

Magicians use well-practiced sleight-of-hand techniques, storytelling, gimmicks, special effects, and surprising events to overwhelm the audience's sensory, cognitive, and even emotional processes. The magician's hand and body movements, narration, and timing can appear innocent, but they influence the facts and events that spectators perceive—and the ones they ignore (see Figure 2). For example, by asking the audience a simple question, a magician can turn their attention inward, generating an internal dialogue that prevents observers from focusing on the actions that take place onstage [35,36]. Similarly, a magician might use humor or other emotive content to misdirect the audience [16,35]. Or a magician might perform more than one gesture at the same time (i.e., "a big motion covers a small motion") to hide a specific action or to split the audience's attention across several spatial locations [7,11,16].

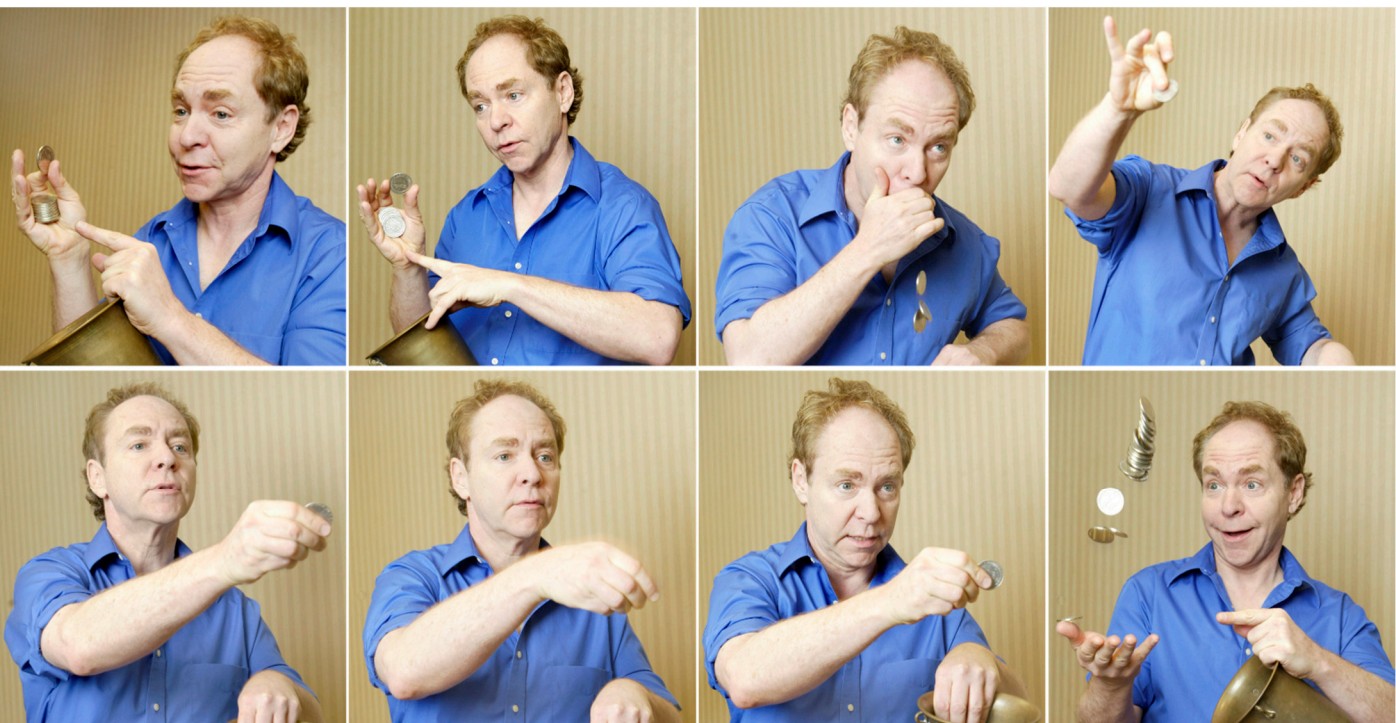

**Figure 2.** Magician Teller's performance of The Miser's Dream trick at the 2007 "Magic of Consciousness" Symposium [37]. The Miser's Dream is a classic magic routine in which myriad coins seem to appear where previously there were none. The magician uses auditory cues with precise timing to make audiences believe that many coins have dropped into the metal bucket when that was not the case. Teller relies on the audience's multi-sensory integration processes, using his body language, gaze direction, and sleight of hand techniques to misdirect spectators. From [16].

The human attentional system gets overwhelmed more easily than we might think. Objects and events that we usually recognize without delay become much more difficult to identify in certain contexts [38–41]. For example, items or patterns can blend perceptually with adjacent ones and become unrecognizable, a phenomenon known as "visual crowding" [42,43]. Similarly, sounds can become unrecognizable when preceded and followed by nonoverlapping noise, a phenomenon known as "auditory masking" [41,44].

Many magic tricks involve multiple cards, coins, etc., changing position, and often rely on the spectators' powerlessness to multitask—that is, their inability to track the locations of such objects and the magician's hands simultaneously. As a result, the audience is effectively prevented from focusing on the elements that may be key to the magical effect. One example is the classic "Cups and Balls" trick, in which the magician uses sleight-of-hand to make balls appear and disappear inside of upside-down cups [7]. The magic duo

Penn & Teller perform this trick with transparent (rather than opaque) cups. Thus, when Penn & Teller place balls secretly inside of the cups, they should be fully visible to audience members. Instead, the balls typically go unnoticed at first, because the spectators' attention is directed elsewhere [11].

The failure to notice the balls as they are secretly loaded into the transparent cups is a kind of change blindness. With change blindness, observers fail to notice that something has become different from how it was before, even if the change is dramatic or if they are looking directly at the location of the changes [45–49]. Moreover, even when such changes are initially noticed, memory limitations can prevent viewers from remembering the specific alterations that took place over a period of time. Magicians sometimes exploit change blindness when swapping out one card for another, to later reveal that the suit or number of the card has changed. Certain objects can be placed in conspicuous locations or removed from sight—all without spectators noticing.

Social media has dramatically increased the volume of data that we must filter, process, and comprehend on an unremitting basis, which may facilitate a comparable kind of deception. Even within a very limited time window, social media displays an overwhelming amount of data to the user—in the form of image, video, and text. All of this takes place at a rapid rate on a small screen, while individuals continue to navigate their daily life, processing events around them, and responding to personal and work demands. The swiftly switching news cycle may similarly prevent viewers from noticing shifts in the framing of a story. In other words, information can come at viewers too quickly for them to stay on top of important details or even track what the latest crisis is. Forced to multitask, people are rendered unable to separate the grain from the straw (in information processing terms, the signal from the noise). As a result, they fail to notice important changes or vital elements in a story.

Visualizing data in animated or video form—showing multiple moving data points—might make people even more vulnerable to misinformation than when they view static images. That is, dynamic images can cause people to miss important changes in the data, such as when attending to one set of data points causes changes in other datapoints to go unnoticed [50]. Research has shown that observers can only track accurately the motion of up to roughly four simultaneously moving objects [51,52].

Insufficient Time to Verify Facts

Magic tricks require that the secret method remains hidden behind the magical effect, creating the illusion that there is no plausible explanation other than that the impossible has happened [53,54]. As such, magicians aim to prevent audiences from carefully evaluating alternative (i.e., veridical) explanations. One way to achieve this is by moving the show at a fast pace, so that there is no time for spectators to deeply analyze the events occurring onstage. Many magical effects would fail if audiences could stop and carefully check for the accuracy of their perceptions (in fact, viewers of magic tricks on YouTube often slow down, pause, and rewind videos with the goal of unveiling magicians' secret methods).

Fact-checking techniques could help counter misinformation, but just like spectators of magic shows, social media users may find themselves unable to conduct in-depth examinations of the content they encounter. Thus, users tend to spend little time evaluating possible explanations, and they rarely interrupt their actions to fact-check online stories [55,56]. This is especially true for groups of people with relatively little practice in online information use, such as older, low-income adults [57,58]. But even those who are fluent in verification practices do not always take the time to authenticate messages and news received on social media. This is all the more difficult when multitasking: indeed, habitual media multitaskers can be worse at judging the accuracy of news. In the end, rather than taking the time to compare data and opinions across reliable sources, social media users tend to simply decide, in the moment, whether a statement is fact or falsehood.

### 2.2. Unattended Events Are Ignored

Events and viewpoints that are presented but are not the focus of attention frequently go unnoticed, a phenomenon known as "inattentional blindness" [59]. Inattentional blindness can happen even for seemingly obvious and salient events: in the classic example, observers watching a video of people passing a basketball failed to see a gorilla walk into the scene and thump its chest [59]. Even viewers who looked directly at the gorilla often did not see it, so long as their attention was directed towards other ongoing events in the video (i.e., counting the number of times the basketball was passed between the players) [60].

Inattentional blindness can be intentionally created by "misdirecting" observers, purposely diverting their attention away from certain events which would be otherwise apparent (i.e., the intruding gorilla in the example above) and towards competing concurrent events (i.e., the players passing the ball to one another) [16,19,61–65]. In magic, misdirection can be so effective that spectators fail to notice events even when looking directly at them [66]. The sophisticated technique by which magicians can draw spectators' attention away from a secret action without redirecting their gaze has been described as "covert misdirection" [16].

Magicians also use narrative and storytelling (i.e., "patter") to distract audiences at crucial points in the performance. Narrative can moreover hold an audience's attention on specifically intended information [61] and thus serve to turn their attention away from other content which may be more important. Just as emotion can drive attention during magic shows [16], compelling and emotional news stories can pull attention towards a narrow slice of information and away from other facts or perspectives.

To sum up, news stories can use misdirection to focus attention toward particular viewpoints or aspects of a report—and therefore away from others—thereby causing potentially important information to be missed. In science reporting, misleading uses of narrative can take a variety of different forms. Narrative can misdirect attention away from questions of whether or not a concept is legitimate, a tactic sometimes referred to as "diversionary reframing" [67]. For example, someone might avoid mentioning allegations made against them, while blaming their past actions and positions on problematic policies or regulations (or pointing to the opposition from another party to avoid taking blame for a political failure). Diversionary reframing tactics include diverting attention away from a controversial statement by undermining the legitimacy of critics [67]. Other narratives reframe discussions or "put spin" on a story by focusing audiences' attention on its positive aspects. For instance, discussions surrounding controversial immigrant detention centers have sometimes focused on the education support provided to detained children to discourage questions about the legitimacy and appropriateness of holding children in such facilities [68].

Even in original scientific articles where discoveries are first announced, scientists frequently report findings in ways that fail to accurately reflect them or their significance [69]. Storytelling is an important part of writing an academic article, as scientists must select the data points, analyses, and all other aspects of a study to present to the reader. Namely, the title, abstract, and conclusions of a research report can all be framed to highlight a particular point, which is acceptable practice in scientific communication. However, storytelling becomes problematic when scientists craft a misleading narrative, with unwarranted interpretations or conclusions that extend beyond the data. Examples include reports that focus on the clinical or societal importance of the findings, even when such results are statistically insignificant [70–72]. In clinical trials, a report with spin might be framed to convey that a treatment is beneficial, despite little or no objective evidence in support of that conclusion [73]. Such spin is prevalent in scientific publishing and reporting [71,74,75], leading to a substantial distortion of research results. This is particularly concerning for clinical trials, where misled patients might proceed with therapies or interventions that are unsupported by science.

Discoveries can also be given "spin" through misleading press releases or the ensuing news reports. Press releases often contain exaggerated causal claims and unsupported

advice [76], while failing to incorporate key limitations or important facts [77–79]. Spin of this kind tends to spread from source to source: when the abstract of the original published article has spin, usually the press release has it too [80]. Similarly, exaggerated claims in press releases tend to be repeated in news reports [76,79–81]. Even if there is no spin in the original article, reporters and editors have incentives to report findings in misleading ways. For example, the pressure to drive clicks to a news website, or to encourage potential readers to subscribe, can lead journalists to exaggerate scientific findings.

On social media, infographics—which are more about storytelling than about visualizing data [82]–and other data visualizations can also facilitate deceptive narratives. Data visualizations can be misleading (intentionally or otherwise) even when they include all the relevant information: for example, misleading labels or distorted axes can cause people to misinterpret data [83]. Whether in news reports or in social media posts, data plots and figures must be read and actively interpreted before their patterns can be understood. Unfortunately, people often form quick initial impressions from visualizations before they carefully examine specific aspects of the images. As a result, an individual that is focusing on the wrong elements of a visualization can miss even highly salient information. In one dramatic example, 93% of people failed to notice a conspicuous image of a dinosaur embedded in visually presented data because they were focusing their attention on another aspect of the plot (the position of the Xs, rather than the Os that formed the dinosaur; [84])—see Figure 3. Misleading captions or content in a Twitter thread or other social media posts could produce comparable attentional failures by directing the viewer's attention towards some patterns and away from others.

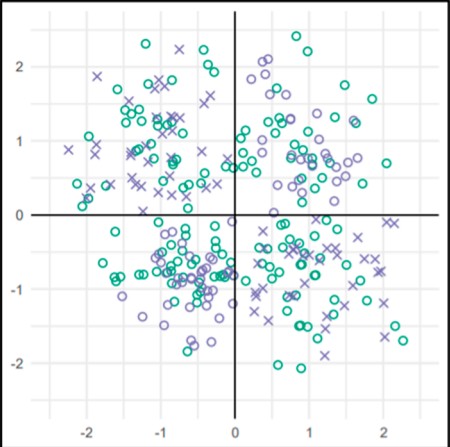 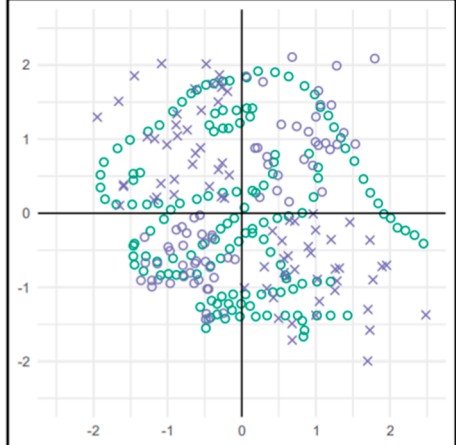

**Figure 3.** In Boger et al. [84], participants viewed a series of plots while they performed a task that required them to focus on one aspect of each display. In one example, participants were tasked with identifying which quadrants of the plots contained blue Xs (i.e., upper left and lower right in the left panel). In some instances, other symbols on the plot formed the contour of a dinosaur (right panel). 93% of participants failed to notice any unusual patterns in the data. When explicitly asked if they had noticed a dinosaur, they reported that they had not. From [84].

Readers can also be misled into incorrect conclusions by confusingly complicated plots, or by peculiar choices in how data is presented (such as by showing the number of COVID-19 cases over time in a pattern of decreasing values, rather than chronologically cumulative case numbers) [50,83,85,86]. People often incorrectly interpret complex information or events as though they were their typical, more simple versions [87]: in magic, for example, spectators often fail to realize that the magician may not cut a deck of cards in the normal way, but using instead a modified maneuver that allows them to control which card ends up on top [88].

Removing or replacing the original narrative—for instance, by posting images without their proper context (i.e., the title, caption, and data labels that belong to an image) can also mislead viewers. In 2020, a data visualization of the fires in Australia was shared online with the inaccurate caption "This is a NASA photograph". In truth, the image

was an artistic visualization, using composite data compiled over time (meaning that the depicted fires were not all simultaneously burning), and somewhat exaggerating the scale of the fires due to glow effects from the way the hotspots were rendered [89]—see Figure 4. Such artistic visualizations are easily mistaken for photographs when reposted in misleading contexts: news media often showed this particular image alongside actual photographs, providing a context that may have further led readers to conclude the image was a photograph too [90].

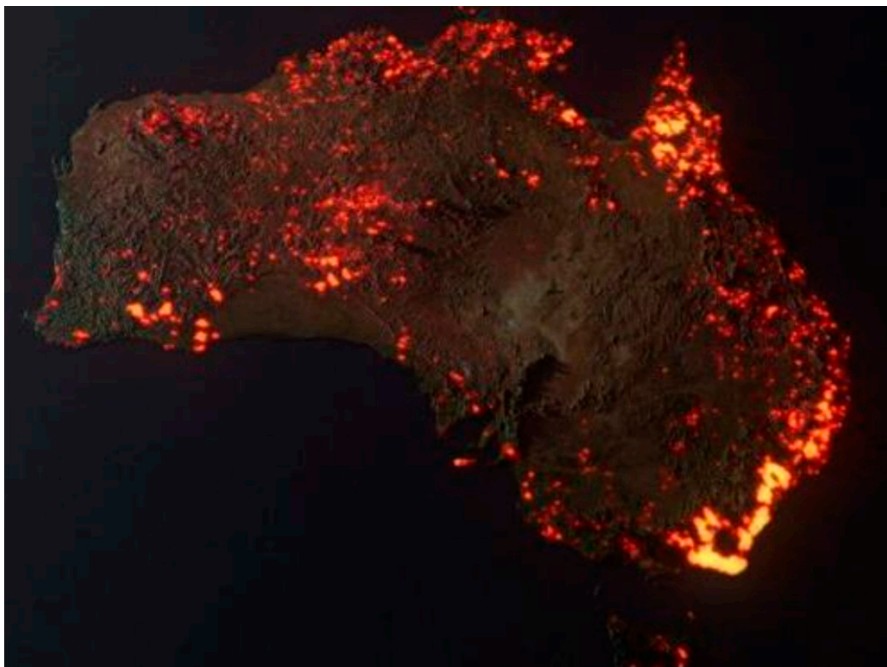

**Figure 4.** This visualization of fire hotspots in Australia may appear lifelike, but it is not a photograph and does not accurately reflect the size or number of fires burning at any one time. Australia is Burning/A 3D Visualization (2019) (available at https://anthonyhearsey.com/australia-is-burning-a-3d-visualisation, accessed on 24 August 2022) by Anthony Hearsey. Reproduction courtesy of the artist.

*2.3. We See What We Expect to See*

Even when there is no intended deception in a report, readers can miss important information. One well-studied reason for these failures is "confirmation bias", in which people irrationally consider only information supporting their expectations [91,92]. Magicians often perform movements that resemble typical everyday actions, relying on confirmation bias to prevent the audience from realizing that something different is happening. The audience may believe that a hand gesture is a common, ordinary action, but that movement may conceal a palmed coin [9,10,16].

People are typically unaware that confirmation bias influences their judgments [93]. Instead, when presented with information that contradicts our beliefs, we often react in a defensive manner. Conversely, we feel more inclined to trust information that is consistent with our prior beliefs than we do other information [91,92,94]. We also assign more relevance to stories that speak from our own perspective, rather than from a different viewpoint. At least in part, confirmation bias may result from familiar information being easier to process than unfamiliar information [95]. Additionally, when people feel motivated to confirm their existing opinions, they choose to interpret information in ways that allow them to feel validated [96,97], relying on the cognitive processes that are most likely to lead to their preferred conclusion [98].

Because social media feeds keep users engaged by tailoring content algorithmically to their preferences, people are relatively unlikely to be exposed to differing viewpoints online [99,100]. Thus, Facebook users tend to see information that supports their existing

beliefs [101,102], which may reinforce prior biases and increase the spread of disinformation [103,104]. The result can be an "echo chamber", where mutual followers tend to have similar opinions that reinforce existing views [105].

Lack of exposure to differing viewpoints cannot be attributed solely to algorithms, however: even when the social media posts challenging users' opinions are visible, that information is often ignored [106] and therefore not shared with other users. In fact, when confronted with too much new or uncomfortable information on their social media feeds, people can feel overwhelmed, reaching a state of "social media fatigue" that may lead them to disengage or temporarily stop using social media [107].

## 3. Memory Illusions

The way we remember the past can substantially deviate from reality. When biographical or historical events are described in a certain way, our recollection of such (biased) description can overshadow our perception of what happened right in front of us.

Studies have shown that memories are most vulnerable to alteration at the time of retrieval. Although the underlying mechanisms are not well understood, the research evidence indicates that retrieving a memory can enhance the subsequent learning of new information [108,109], and change the original memory [110,111]. That is, whenever we relive a memory, we re-store it as a new memory for later access—essentially wiping off the old memory in the process.

Our memory's malleability provides a fertile ground for misinformation. Magicians sometimes take advantage of this pliability by recasting events that took place onstage in a manner that will bias spectators' memories of the performance [19] and prevent them from figuring out the secret method when they try to puzzle it out later [7]. This same process may explain the enhanced susceptibility of eye witnesses in criminal proceedings to misleading suggestions [109].

*The Illusory Truth Effect: Repeated Lies Become Plausible*

Events and stories seem truer when repeated multiple times. For example, if a magician tosses a coin from hand to hand a few times, and then fakes a throw from right hand to left hand (while secretly concealing the coin in the right hand), it can look as though the magician catches the tossed coin in their left hand. Then, when the magician opens that hand and reveals it to be empty, the coin appears to have vanished [9]. Similar tricks can be performed with balls [62,112] or other items: the object is truly thrown a few times and then the magician merely pretends to throw the object. The initial repetitions create the expectation and false perception that the same event is happening again. This illusion results in part from "priming", in which the presentation of a stimulus makes subsequent appearances of the stimulus more salient [113].

Repetition can also prime people to believe false information. News headlines that are seen only once are more likely to be believed if they are true [114,115]. But news headlines that are seen multiple times are more likely to be believed *irrespective of their accuracy*. That is, a repeated message becomes increasingly plausible each time around, even if people initially believed it to be false [115,116]. One possible explanation is that repeated exposure makes information easier to process. Thus, people can interpret their feeling of ease with a message as evidence of its veracity [117]. This phenomenon, called the "illusory truth effect", can last months after exposure [118].

Although highly implausible statements (such as "a single elephant weighs less than a single ant") continue to be considered false in the face of repetition [5,6], research has shown that even such implausible statements seem somewhat more plausible when repeated [115]. The illusory truth effect persists when individuals are warned about its existence [119], and when the message contradicts prior knowledge [120,121]. In other words, even if people are able to identify the truth, repetition can boost the perceived accuracy of false statements [121]. Provided enough repetition, obvious lies may begin to seem credible.

Labels identifying material as potentially false can reduce the size of the illusory truth effect in some cases [122], but they do not eliminate it. With repetition, inaccurate statements become more credible even when valid alternatives are available [123], and even when the misinformation comes from a source that is known to be untrustworthy and unreliable [124]. Thus, an expert in a field repudiating a piece of news as false may not be enough to counter repeated misinformation.

In the past decade, there was some concern that repeating a story to refute it could backfire. The reasoning was that (1) people would dismiss any new information that they disagreed with, due to confirmation bias, and (2) additional exposure to false information (even if it was immediately refuted) might strengthen its hold on people's minds [101,125–127]. Nevertheless, such "backfire effects" have proven to be rare, and there is evidence that direct refutations can be successful in correcting misinformation [128–131].

## 4. Illusions of Choice

The way we think about and remember events can permanently alter our understanding of them [132–134]. Many factors in our environment can alter our behavior, and we tend to assume that what we see is as it appears—that there are no concealed events secretly changing the outcome of an action. The result can be an "illusion of choice", in which we believe we are making decisions freely, but specific results are inevitable and out of our control.

### 4.1. Our Decisions Can Be Influenced

If a magician asks you to choose a card, it may seem as though you are free to make any choice you like. However, magicians have a wide variety of "forcing" techniques that constrain spectators' choices, making them much less free than they appear to be [135]. That is, the magician influences or limits an individual's choice without their awareness [136,137]. Depending on the technique used, the spectator's choice may be completely predetermined, or the magician might merely enhance the probability that they will make a particular decision. In one simple example, the magician might discretely or overtly discard any card chosen by an audience member—possibly with an entertaining flourish—until the "correct" card is selected. Alternatively, the magician may use gestural and conversational primes to encourage the spectator to make a certain choice (such as drawing the number three in the air and making a diamond shape with their hands to increase the chance that a volunteer will pick the three of diamonds [138]). Time pressure to produce an action quickly (such as picking a card) can urge audiences to act without thinking deeply, and thus compel them to perform the most obvious action [54]. Therefore, a magician can force a spectator's card selection by having them quickly choose a card at a time when only one card is visible [54]. Or a magician might ask the spectator to draw the card from the top of a shuffled deck—a card whose placement they have controlled with sleight of hand. The result is that audiences believe they are freely making their own decisions when that is not the case. Alternatively, a spectator's choice could be free but irrelevant to the outcome [139], as when either selection will lead to the same end result (a magic technique known as "magician's choice" [140]). The conclusion is predetermined, and the spectator's involvement in the outcome is an illusion.

Social media users may choose to like some messages and not others, but the platform may be showing them a small subset of posts selected by an algorithm. If users are presented only with information that supports their existing attitudes, they will never have the actual option to accept or reject contrary opinions.

The phenomenon known as "choice blindness" also shows that people do not only misremember manipulated choices as their own, but they also vigorously defend choices that they did not make (though they believe they did) [141].

One example in which a magician can exploit this kind of vulnerability is by covertly changing the card chosen by a volunteer and then asking them to "go ahead, take your card" [88]. Comparable bait-and-switch tactics can be found in online advertising: some

ads are designed to look like real news articles, opinion polls, or online search results. The nature of such ads is misleading in itself, with many users unable to reliably identify the content as advertisements [142]. Consequently, people click on ads believing that they are moving on to another trustworthy story on the same website [143], but instead they link to external sites that gather their email addresses, or help to financially support misinformation websites [143,144]. Because these ads often use unsubstantiated language or frame information in misleading ways, they can also spread misinformation directly [144,145]. Bait-and-switch tactics are also common when using search engines: Google, for example, has increasingly dedicated space to advertisements that look like search results [146,147].

### 4.2. Misleading or Unbalanced Viewpoints

Magicians also manipulate audiences' interpretation of events by controlling their viewpoint. Magical effects that seem wondrous from the vantage point of spectators can appear mundane when viewed from another angle. Thus, in stage performances the perspective is often controlled by the context: the audience can only view the trick from the front of the stage. Movements, objects, and events that are solely visible from backstage are concealed from the audience's view.

In a news story, how different kinds of information or viewpoints are presented can similarly lead audiences to accept a single interpretation of facts as truthful, without ever knowing that alternative perspectives existed.

Even when several perspectives are represented, news can give imbalanced treatments to different viewpoints. Journalists typically value balanced reporting, and strive to present information accurately [148], but in practice there is no widely recognized and agreed upon definition of balanced reporting [149]. That is, balanced reporting might refer to seeking out and presenting all different sides or angles in every story, or to approaching information in a detached and neutral way. Regardless of the chosen approach, finding an appropriate balance is more difficult than one might assume.

Other times, stories simply omit information and thereby prevent readers from considering it. For instance, a story might mention a change to a number (e.g., "5% fewer COVID-related deaths") rather than the actual number (which, although lower than before, might still be strikingly high).

Just as spectators in a magic show lack the information or the time to figure out the secret method behind a magical effect, reporters may be unable to determine fact from fiction for comparable reasons: a scarcity of time, scientific expertise, or other resources. Local news sources have especially struggled in the past decades [150–153] and may need to reduce resources to cut costs, possibly including cutting fact-checking verification practices.

Unlike magic audiences, journalists are expected to extract (and provide) veridical information from potentially deceitful appearances. Yet, with a limited ability to determine the accuracy of two positions, some journalists seeking to avoid perceived bias will present both sides of a dispute with equal weight, irrespective of their respective merits [154,155]. Unfortunately, if one of such viewpoints is based on untruths and misinformation, portraying both sides as equally valid is inherently misleading. Likewise, giving domain experts and nonexperts an equal voice can distort the truth. For example, journalists might misrepresent the current scientific consensus on climate change by quoting both environmental scientists and climate change deniers [155–158].

If 1% of scientists hold a particular viewpoint but they receive an equal treatment to the other 99%, such reporting would give the incorrect impression of a 50/50 split in opinion. Because audiences do not know the actual distribution of opinions among scientists, equal balance can produce a form of sampling bias, where minority opinion is given too much weight and misinformation appears to be legitimized. Equal balance practices may also increase public distrust of science: when scientific findings are presented alongside misinformed criticisms, the scientific conclusions can appear less certain than they really are. This situation becomes even more problematic when there are more than two

sides in a story: given time and space constraints in reporting, journalists can experience difficulties when selecting which viewpoints to include, omit, or discuss in detail.

Overall, the above kind of "balanced" reporting can undermine the interests of audiences who try to draw meaningful conclusions, both by highlighting misinformation and by limiting the time given to in-depth discussions of accurate information [159,160]. One study found that stories with an equitable tone (toward two political candidates) were less likely to focus in-depth on substance or issues, as well as less likely to include sourced content. They were also generally shorter in length than "slanted" stories that favored one candidate over the other [161]. Whereas "balanced" reporting practices may be expeditious to journalists, giving equal importance to truth and to misinformation clouds matters and leads to information biases [148].

### 4.3. Why Warnings Fail to Prevent Misperceptions

The sense of wonder that accompanies a magic show requires that the performance seems impossible. But being warned that the magical effects are "just tricks" may do nothing to diminish the emotions felt by the audience. The warning is, in fact, redundant: audiences typically know that magic shows are not accurate portrayals of physical reality.

In false news reports, misinformation similarly triggers emotional responses even in contexts where it should not. When misinformation is labelled as potentially inaccurate and contested (e.g., Facebook stating on a page that independent fact-checkers have said the information is false, or Twitter warning users that claims about election fraud are disputed), readers often still believe it and share the misleading claims [115,162–166]. In fact, even when readers notice the warnings, they often ignore them, so long as the warning does not interrupt the readers' actions [163,167,168]. Even if warnings are heeded, the fact that only some messages include them can make people think that the other messages that do not include such warnings must be accurate—when they may contain other misinformation that the platform failed to detect [163].

## 5. Potential Answers and Areas for Future Research

Many of the cognitive biases discussed above are ingrained and therefore very difficult to circumvent or prevent. However, being aware of these vulnerabilities may help individuals to critically evaluate information and ameliorate bias, as well as assist developers and legislators in the creation of more effective tools for combatting misinformation. Topical experts can be encouraged to vocally dispute misinformation, and scientists supported to engage in communication and outreach practices, to help journalists and the public to accurately understand discoveries. Financial assistance can be provided to local news outlets to provide them with fact-checking resources.

Some direct steps can also be taken to decrease the flow of misinformation, including the implementation of warnings on disputed messages. Warnings are often, but not always ignored, and therefore can work to slow the spread of misinformation [163,167,168]. Further, warnings can be made more effective by interrupting a user's actions (e.g., requiring a mouse click to continue) and by periodically changing so that users do not dismiss them or bypass them out of habit. At the extreme, users who spread large amounts of misinformation can be deplatformed, removing their messaging from a social media platform. An important caveat is that as measures against misinformation grow in severity, so will the likelihood that they will be viewed as paternalistic, imposing, punitive, and restrictive of free speech [169,170].

Some spectators of magic shows enjoy focusing their efforts on dissecting tricks and identifying the methods behind magical effects. Similarly, some social media users take effortful steps to authenticate dubious claims, helping to slow the spread of misinformation with fact-checking behaviors. Users who share their findings on social media can encourage others to remove misleading messages or to reconsider their behavior when posting in the future. These efforts by individual users should be supported, and their fact-checking encouraged, by providing easy-to-use and freely available verification tools. Some fact-

checking resources have become widely available and used, such as the website Snopes.com. Similar initiatives should be supported, with the caveat that users should also be educated about which tools are reliable, given that sites *masquerading* as fact-checking services have also proliferated. Fact checking, verification, and authentication processes that are already in place should also be revisited regularly to consider new misinformation techniques and tools, particularly when dealing with topics where misinformation can create substantial harm.

Engineers and other developers should take misuse-related considerations seriously when determining research priorities and when creating or modifying tools such as social media platforms. Developers working with such tools should proactively act to minimize foreseeable harmful applications. Domain experts in neuroscience and psychology should be included—and the factors discussed above considered—in discussions of these issues and in the development of legislation dealing with misinformation.

False news has existed for a long time, and it will continue to exist despite efforts to counter it. As communication becomes progressively faster and easier, misinformation will also increase in scale. With technological advances, intentional disinformation attempts will likely become more effective, and new methods and approaches for deceiving the public will appear. For example, artificial intelligence (AI) may enable new kinds of automated disinformation attempts at ever-larger scales. Misinformation will also expand on smaller, personal levels, as technologies enable greater numbers of users to create authentic-seeming fake images and videos. These images can be employed for image-based abuse (also called "revenge porn"), bullying, blackmail, political sabotage, and other malicious purposes. Fake images might even be used as false evidence of a crime, potentially leading to wrongful convictions [171].

Individuals can now use AI tools to produce "deepfake" videos that appear convincingly real, in which new words are digitally inserted into audio tracks, or the face or body of a person is digitally altered to match someone else's appearance [172,173]. Deepfake videos will likely become increasingly realistic as software tools improve. Recently developed text-to-image synthesis techniques can also generate new—and often realistic—images from visual descriptions provided by the user as text [174]. As a result, users can very quickly and easily create new images to include in misleading social media messages. There is no equivalent in magic to date: although magicians often incorporate cutting-edge technologies into their acts, no existing automated tool has yet been used to create magical effects from simple text prompts. Though research on magic does not yet illuminate the impact of—or what tools may combat—AI-facilitated deceptions, we expect that determining how people perceive the accuracy or falsehood of audio and video recordings will be a necessary direction for future research on misinformation.

## 6. Conclusions

Magical effects are incredibly robust: they work even though audiences know that they are being tricked. Similarly, people often accept and disperse misinformation despite warnings that the facts are disputed and potentially false [115,162–166]. Thus, increasing the awareness of scientific facts has proven ineffective in countering the flow of misinformation [175,176]. In news stories, as in magic performances, people do not view information in a detached and neutral way [177]. Psychological and emotional appeal can be much more important than factual accuracy. In this paper, we reviewed some factors that help explain how people react to misinformation, grounding the discussion around the ways in which magicians exploit audiences' vulnerabilities and biases in perception and cognition.

Audiences are often fooled by the concealment of key information in magic shows: no spectator can accurately understand data that are absent. Likewise, due to reframing and spin, and even outright data falsification, news reports and social media posts are often impossible to evaluate accurately by readers.

Even when information is present—and highly salient—observers can fail to process it accurately. Attention can be misdirected away from central facts and data. Confirmation

bias and other cognitive biases can cause people to miss or disregard accurate information. Sampling biases can mislead news story consumers and social media users by granting legitimacy to fringe opinions. Repetition causes false information to gain plausibility with each iteration.

Research studies on perception and cognition have been—and continue to be—conducted to determine the situations in which magic tricks succeed or fail. Scientists may draw on this growing literature to help identify the scenarios that withstand misinformation, and to devise more fruitful methods for countering its spread. Reducing the prevalence and societal impact of misinformation will require multi-layered approaches that consider the range of biases we discussed here.

**Author Contributions:** Conceptualization, R.G.A. and S.M.-C.; Writing—Original Draft Preparation, R.G.A.; Writing—Review and Editing, R.G.A., S.L.M. and S.M.-C.; Funding Acquisition, S.L.M. and S.M.-C. All authors have read and agreed to the published version of the manuscript.

**Funding:** This work was supported by the New York State Empire Innovator Program, by the National Science Foundation (Award 1734887 to S.M.-C. and S.L.M.; Award 1523614 to S.L.M.), and by the National Institute of Health (Awards R01EY031971 and R01CA258021 to S.M.-C. and S.L.M.).

**Data Availability Statement:** No new data were created or analyzed in this study. Data sharing is not applicable to this article.

**Conflicts of Interest:** The authors declare no conflict of interest. The funding sponsors had no role in the design of the study; in the collection, analyses, or interpretation of data; in the writing of the manuscript; or in the decision to publish the results.

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
