# Peer review of "What the Neuroscience and Psychology of Magic Reveal about Misinformation"

_publications, doi:10.3390/publications10040033_

Round 1

Reviewer 1 Report

It is a narrative review paper, in the style of a clinical review, in this case applied to the experience of misinformation

It has a clearly defined goal. It is very clearly written, effortlessly understood, concise, precise and in straightforward language. The references are current and varied, consistent with the proposed work.

Despite the greater freedom in the structure of a work of these characteristics, it is suggested as necessary to include a brief specific section on methodology, in which the criteria for selecting the common factors between the experience of magic during a performance and misinformation.

I consider it to be an original work, which brings a very new and enriching perspective to the study of misinformation, relevant and well justified, which advances knowledge in its area of ​​study. With a good research approach. At the same time, with a clearly constructive spirit, a minor modification is proposed, the inclusion of a methodological section, which I consider can reinforce the work.

Author Response

Reviewer 1:

We thank the reviewer for their kind words (“It has a clearly defined goal. It is very clearly written, effortlessly understood, concise, precise and in straightforward language. The references are current and varied, consistent with the proposed work.”) and (“I consider it to be an original work, which brings a very new and enriching perspective to the study of misinformation, relevant and well justified, which advances knowledge in its area of ​​study. With a good research approach.”) and for the helpful suggestions for improving this manuscript.

Despite the greater freedom in the structure of a work of these characteristics, it is suggested as necessary to include a brief specific section on methodology, in which the criteria for selecting the common factors between the experience of magic during a performance and misinformation…. a minor modification is proposed, the inclusion of a methodological section, which I consider can reinforce the work.

As the reviewer points out, this narrative review relies on a broader and more free approach than a systematic review. We selected articles not only through extensive key word searches and searches of the references of retrieved articles, but also through discussions with experts in the fields of neuroscience and magic, as well as from the authors’ personal experience at the intersection of both domains. The information included in the article is built from the articles we referenced. We do not feel that the addition of a methods section would serve to delineate these rather loose criteria for selection/inclusion/exclusion of material. Having said this, we will defer to the editor’s judgement on this matter and include a methods section if the editor advises.

Reviewer 2 Report

This review draws connections between the psychology/neuroscience of magic and individuals' interpretations of disinformation online. The authors provide a summary of magic-related research in both fields and give specific examples of how these concepts relate to the public’s consumption of online news. Overall, the paper is well-written and easy to follow, and the authors’ idea to connect these seemingly unrelated concepts is novel. As a fan of psychology, social media, AND magic tricks, I thoroughly enjoyed reading the article.

I have 2 suggestions for somewhat significant changes to the manuscript:

1)   I think that including neuroscience but not psychology in the title is misleading. The examples included and articles cited seem equally focused on neuroscience AND psychology. Adding psychology to the title may also increase readership.

2)   My other major suggestion is to add a section on potential solutions and/or areas for future research. The authors make it clear that there are issues with how people process information they see online and connect those problems to magicians’ techniques, but do not offer any solutions. This section wouldn’t need to be extensive, but its inclusion would demonstrate that potential solutions exist.

Next, I have a series of minor suggestions.

1)   The authors briefly mention “crowding” and “auditory masking” in lines 83-86 but do not offer definitions. An explanation and brief example of both terms would enhance understanding.

2)   On line 88, change “incompetence” to “inability”

3)   The paragraph that begins on line 118 requires more clarification. Specifically, I’m not sure what is meant by “Certain kinds of presentations can be especially problematic.” My guess is that the authors are claiming that the method in which information is shared or presented can impact how it is perceived. I suggest rewording that first sentence and adding a bit more explanation to the paragraph. Another example may be helpful.

4)   In the paragraph on inattentional blindness beginning on line 149, a summary of the gorilla video is required. I’m familiar with the video, but don’t assume that all readers will be. Provide a brief description of the video and an explanation of why observers fail to notice the gorilla.

5)   Provide a definition for “covert misdirection” on line 160. It’s implied that this type of misdirection is unique to magicians and particularly effective, so explain what differentiates it.

6)   I suggest using a more relevant example than “publish or perish” in lines 175-176. Because this article focuses on online news stories, a hypothetical example about a political scandal or something similar seems more relevant.

7)   I think a transition is required at the beginning of the paragraph on line 183. You jumped from news stories to scientific articles, so I wasn’t sure if you meant that scientists themselves put a spin on their own discoveries or if the spin is added by journalists who are reporting on the discoveries. It becomes clear later in the paragraph that you are talking about scientists writing peer-reviewed articles, but a transition would improve the flow of the argument.

8)   The paragraph in lines 198-204 would benefit from a brief explanation on why journalists often report scientific findings in misleading ways. (The pressure to publish online quickly, headlines that encourage clicks, etc.)

9)   I suggest adding a discussion of Echo Chambers and Motivated Reasoning to section 2.3.

10)                Add more background information in the introductory paragraph of section 4 (lines 322-326). Provide an explanation of what you mean by “Illusions of choice” before giving examples

11)                Examples would enhance your explanation of choice blindness in lines 352-359. The magic example is good, but can you provide an example related to online information/misinformation?

12)                I would also discuss the role of algorithms in search engine results in the paragraph lines 360-368. For example, how Google shows you the results you are most likely to click rather than the results that are most relevant.

13)                Section 4.2 would benefit from more connection to magic throughout. In every other section that authors do an excellent job of continuing to draw connections between magic shows and online information, but in this section magic is only mentioned in the 1st paragraph

14)                In paragraph lines 427-434 I would connect this research to social media platforms’ recent attempts to warn against false information. Facebook and Twitter for example have started flagging these posts, but is that not achieving the desired results?

15)                Line 448, change “what” to “that

Author Response

Reviewer 2:

We thank the reviewer for their kind words (“Overall, the paper is well-written and easy to follow, and the authors’ idea to connect these seemingly unrelated concepts is novel. As a fan of psychology, social media, AND magic tricks, I thoroughly enjoyed reading the article.”) and for the helpful suggestions for improving this manuscript.

I have 2 suggestions for somewhat significant changes to the manuscript:

1)   I think that including neuroscience but not psychology in the title is misleading. The examples included and articles cited seem equally focused on neuroscience AND psychology. Adding psychology to the title may also increase readership.

Done.

2)   My other major suggestion is to add a section on potential solutions and/or areas for future research. The authors make it clear that there are issues with how people process information they see online and connect those problems to magicians’ techniques, but do not offer any solutions. This section wouldn’t need to be extensive, but its inclusion would demonstrate that potential solutions exist.

We now include a section along these lines.

Next, I have a series of minor suggestions.

1)   The authors briefly mention “crowding” and “auditory masking” in lines 83-86 but do not offer definitions. An explanation and brief example of both terms would enhance understanding.

We have clarified these definitions.

2)   On line 88, change “incompetence” to “inability”

Done.

3)   The paragraph that begins on line 118 requires more clarification. Specifically, I’m not sure what is meant by “Certain kinds of presentations can be especially problematic.” My guess is that the authors are claiming that the method in which information is shared or presented can impact how it is perceived. I suggest rewording that first sentence and adding a bit more explanation to the paragraph. Another example may be helpful.

We have clarified the text in this section.

4)   In the paragraph on inattentional blindness beginning on line 149, a summary of the gorilla video is required. I’m familiar with the video, but don’t assume that all readers will be. Provide a brief description of the video and an explanation of why observers fail to notice the gorilla.

Done.

5)   Provide a definition for “covert misdirection” on line 160. It’s implied that this type of misdirection is unique to magicians and particularly effective, so explain what differentiates it.

We now define this term.

6)   I suggest using a more relevant example than “publish or perish” in lines 175-176. Because this article focuses on online news stories, a hypothetical example about a political scandal or something similar seems more relevant.

Done.

7)   I think a transition is required at the beginning of the paragraph on line 183. You jumped from news stories to scientific articles, so I wasn’t sure if you meant that scientists themselves put a spin on their own discoveries or if the spin is added by journalists who are reporting on the discoveries. It becomes clear later in the paragraph that you are talking about scientists writing peer-reviewed articles, but a transition would improve the flow of the argument.

We have improved this transition as suggested.

8)   The paragraph in lines 198-204 would benefit from a brief explanation on why journalists often report scientific findings in misleading ways. (The pressure to publish online quickly, headlines that encourage clicks, etc.)

Done.

9)   I suggest adding a discussion of Echo Chambers and Motivated Reasoning to section 2.3.

We have added a discussion of work related to these topics.

10)                Add more background information in the introductory paragraph of section 4 (lines 322-326). Provide an explanation of what you mean by “Illusions of choice” before giving examples

Done.

11)                Examples would enhance your explanation of choice blindness in lines 352-359. The magic example is good, but can you provide an example related to online information/misinformation?

Done.

12)                I would also discuss the role of algorithms in search engine results in the paragraph lines 360-368. For example, how Google shows you the results you are most likely to click rather than the results that are most relevant.

Done.

13)                Section 4.2 would benefit from more connection to magic throughout. In every other section that authors do an excellent job of continuing to draw connections between magic shows and online information, but in this section magic is only mentioned in the 1st paragraph

We have revised this text to increase the connection with magic performances.

14)                In paragraph lines 427-434 I would connect this research to social media platforms’ recent attempts to warn against false information. Facebook and Twitter for example have started flagging these posts, but is that not achieving the desired results?

Done.

15)                Line 448, change “what” to “that

Done.